# Windswept Deformity a Disease or a Symptom? A Systematic Review on the Aetiologies and Hypotheses of Simultaneous Genu Valgum and Varum in Children

**DOI:** 10.3390/children9050703

**Published:** 2022-05-10

**Authors:** Niels J. Jansen, Romy B. M. Dockx, Adhiambo M. Witlox, Saartje Straetemans, Heleen M. Staal

**Affiliations:** 1Department of Orthopaedic Surgery, Maastricht University Medical Centre (MUMC+), 6229 HX Maastricht, The Netherlands; niels_jig_jans@hotmail.com (N.J.J.); r.dockx@student.maastrichtuniversity.nl (R.B.M.D.); ma.witlox@mumc.nl (A.M.W.); 2Department of Paediatric Endocrinology, Maastricht University Medical Centre (MUMC+), 6229 HX Maastricht, The Netherlands; saartje.straetemans@mumc.nl

**Keywords:** windswept deformity, genu valgum, genu varum, children, rickets

## Abstract

Objective: The objective of this study is to create an overview of the possible aetiologies of windswept deformity and to emphasize the points of attention when presented with a case. Methods: A systematic search according to the PRISMA statement was conducted using PubMed, African Journals Online, Cochrane, Embase, Google Scholar, and Web of Science. Articles investigating the aetiology of windswept deformity at the knee in children, and articles with windswept deformity as an ancillary finding were included. The bibliographic search was limited to English-language articles only. The level of evidence and methodological appraisal were assessed. Results: Forty-five articles discussing the aetiology of windswept deformity were included. A variety of aetiologies can be brought forward. These can be divided into the following groups: ‘Rickets and other metabolic disorders’, ‘skeletal dysplasias and other genetic disorders’, ‘trauma’ and ‘descriptive articles without specific underlying disorder’. With rickets being the largest group. Interestingly, in the group without a specific underlying disorder, all patients were from African descent, being otherwise healthy and presented with windswept deformity between two and three years of age. Conclusion: We have presented an overview that may help identify the underlying disorder in children with windswept deformity. A step-by-step guide for clinicians who see a child with windswept deformity is provided. Even though, according to the Oxford level of evidence, most articles have a low level of evidence.

## 1. Introduction

In 1975, Oyemade made mention of windswept deformity (WSD), under the term “varovalga”, characterised by the formation of a valgus deformity of one knee, and a varus deformity of the contralateral knee [1]. In 1976, the term windswept deformity was used by Fulford et al. to describe the general postural deformity acquired by children with cerebral palsy, in their first weeks of life [2]. Consequently, windswept deformity is used to describe the phenotypical presentation of a varus and valgus deformity, however, the location of this may vary, as well as the underlying pathology. In this article, we will focus on windswept deformity of the knee.

Although the long-term consequences of windswept deformity have not yet been described, it is presumable that it has a similar impact as other angular deformities of the knees. Untreated, it can lead to further deformity, gait abnormalities, limb shortening and osteoarthritis [3]. 

To date, the aetiology of windswept deformity remains unknown. Oyemade and Smyth described windswept deformity of the knee in previously healthy children, mostly of sub-Saharan descent, with normal developmental milestones [1,4]. The onset usually occurs in the second or third year of life, shortly after the onset of walking. According to Smyth, the valgus deformity develops first, rapidly followed by a varus deformity on the contralateral side [4]. Suggested hypotheses can be divided into the following categories: metabolic or dietary [5,6,7,8,9,10], mechanical pressure [1,11], reactive to unilateral disease [12], genetic/race [1,4,13] and traumatic [14]. Windswept deformity can be treated either surgically (corrective osteotomies, stapling) or conservatively (plaster casting) [11]. 

Apart from varovalgum at the knee, different forms of windswept deformity exist, including hip deformity in children with cerebral palsy [15]. The aim of this study is to perform a systematic search according to the PRISMA statement, in order to analyse which aetiologies of windswept deformity have been published, and assess their level and quality of evidence, as well as any bias. Our aim is to create an overview of the possible aetiologies for windswept deformity and emphasize the points of attention when presented with a case.

## 2. Materials and Methods

This systematic review was written according to the PRISMA statement for reporting systematic reviews and meta-analyses [16].

### 2.1. Eligibility Criteria

Types of studies: all articles investigating the aetiology of windswept deformity at the knee and all articles with windswept deformity as an ancillary finding. The language was restricted to articles written in English. There was no restriction on the date of publication. 

Types of participants: all patients with a presentation of windswept deformity during childhood (0–18 years) were included. There were no restrictions on gender or race. 

### 2.2. Information Sources

The following databases were searched on 16/10/2020: PubMed, African Journals Online, Cochrane, Embase, Google Scholar, and Web of Science. Articles were screened for eligibility, based on title, abstract and the full text. Additionally, the reference lists of the included articles were screened for further identification of any relevant articles and they were included where applicable. 

### 2.3. Search Strategies

The following search terms were used. The limit “English” was added as mentioned in the eligibility criteria. 

1.(windswept deformity) OR (windswept);2.(((genu valgum) AND (genu varum)) OR ((genu valgum[MeSH Terms]) AND (genu varum[MeSH Terms]))) OR (combined valgus and varus knee) AND ((humans[Filter]) AND (allchild[Filter])) AND (English[Filter]);3.(((((varo-valgum) OR (varovalgum)) OR (genu varo-valgum)) OR (genu varovalgum)) OR (varo-valga)) OR (varovalga).

An overview of the search can be found in Appendix A. 

### 2.4. Study Selection and Data Extraction

Studies were selected based on the eligibility criteria. First, duplicates were removed, followed by the selection based on title and abstract. The remaining articles were screened for their eligibility based on the full text. The included articles were screened upon a second survey and consensus amongst the authors was reached. The relevant data were extracted and reviewed by the authors.

### 2.5. Level of Evidence and Quality Assessment 

The level of evidence was scored according to the Centre for Evidence-Based Medicine (CEBM) by three authors, followed by an assessment of methodological appraisal. The latter was performed according to the Joanna Briggs Institute (JBI) for case reports, case series, cohort studies, case-control studies, and cross-sectional studies [17]. Cut-off values were obtained using a scoring system, where a “yes” answer scored 2 points, “unclear” 1 point, “no” 0 points, and “not applicable” was subtracted from the maximum obtainable score. For each article, the obtained score was divided by the maximum obtainable score which led to a percentage. An article was considered to be of high quality when the score was 75% or higher, moderate between 50% and 75%, and low when the score was lower than 50%. Articles classified as “literature review” did not meet the criteria for level of evidence scoring or critical appraisal. An overview of the methodological appraisal can be found in Appendix B.

## 3. Results

An overview of the results and the number of records retrieved from the final search can be found in Figure 1.

The three searches, performed in six databases, yielded a total of 773 records. After removing 193 duplicates, 580 records remained. Of these records, 424 were excluded based on title/abstract screening, leaving 156 records to be assessed based on the full text. From these 156 full-text records, 109 were excluded, as shown in Figure 1. This resulted in a total of 47 publications that were selected, of these, all the references (*n* = 1201) were screened from which four additional articles were selected. Upon second survey another six articles were excluded based on age > 18 years (*n* = 1), not being windswept deformity (*n* = 2) and no clear aetiology stated (*n* = 3). This resulted in a final number of 45 selected publications. 

Although 45 articles is a substantial amount of included articles, only very few focus on the aetiology of windswept deformity, with most articles describing windswept deformity as an ancillary finding. The Oxford CEBM level of evidence regarding these publications is generally low, with most articles ranked as level IV. Additionally, the quality of the articles, assessed as described in the methods, varied greatly, ranging from a score of 50% to 100%.

In Table 1, general information can be found about the articles, including the country of study, study design, the aim of the study, and the main information related to windswept deformity, as well as the Oxford CEBM classification on the methodological quality score.

From the selected articles, a variety of aetiologies for windswept deformity can be brought forward. These can be divided into the following groups:Rickets and other metabolic disorders;Skeletal dysplasias and other genetic disorders;Trauma;Descriptive articles without the specific underlying disorder.

**Figure 1 children-09-00703-f001:**
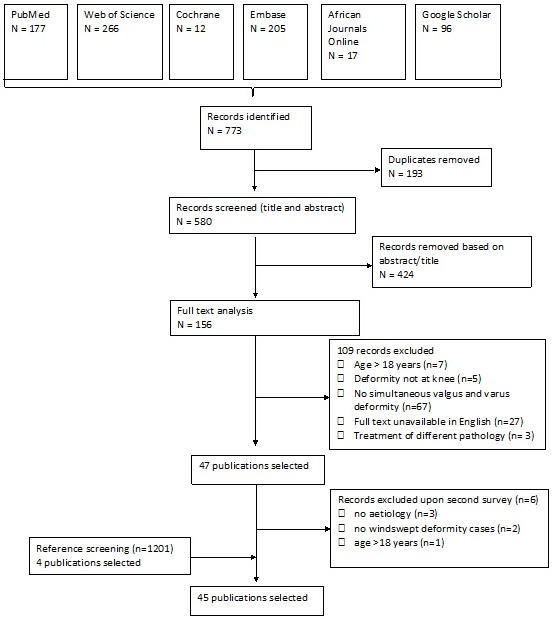
Flow diagram of studies screened and included in the review.

**Table 1 children-09-00703-t001:** Overview of the demographic data of all included studies regarding windswept deformity at the knees.

Article	Country of Study	Study Design	Aim of Study	Elaboration of WSD	WSD Aetiology	Level of Evidence (CEBM) and Methodological Quality
Akpede et al. [5]	Nigeria	Prospective cross-sectional	Determine the prevalence of clinical and biochemical rickets.	Two of ten patients who showed biochemical rickets, though not radiologically, did show WSD, suggesting a form of healed rickets.	Rickets, no radiographic evidence of active rickets	IV high
Al Kaissi et al. [18]	Austria	Case report	WSD in a patient with Schwartz-Jampel syndrome (SJS).	One patient with SJS, with WSD.	SJS	IV high
Al Kaissi et al. [19]	Austria	Case series	Record and discuss WSD in patients with X-linked hypophosphataemic rickets.	In seven patients with hypophosphataemic rickets, the most common angular deformity is WSD.	Hypophosphataemic rickets (from PHEX mutation)	IV moderate
Bar-On et al. [20]	Israel	Retrospective case series	Characterise deformities in patients with renal osteodystrophy (ROD).	One out of five patients showed WSD.	ROD	IV high
Bar-On et al. [14]	Israel	Retrospective case series	To investigate patients with insensitivity to pain.	One patient developed WSD as a consequence of growth disturbance due to untreated fractures of the growth plate.	Trauma	IV high
Bharani et al. [21]	India	Case report	To describe two siblings with sickle cell anaemia, presenting with bilateral lower limb deformities.	Two siblings, both male, 2 and 10 years old with progressive genu valgus on the right, genu varus on the left.	Distal renal tubular acidosis (dRTA)	IV High
Bhimma et al. [22]	Natal, South Africa	Case series	To determine the clinical spectrum of rickets among black children.	WSD was found in two patients with vitamin D deficiency and one patient with Ca deficiency in a total population of 37 patients.	Vitamin D and/or Ca deficiency rickets	IV high
Dudkiewicz et al. [23]	Israel	Case report	Describe the procedure of bone elongation in hypophosphataemic rickets.	One WSD with right genu valgum/left genu varum.	Hypophosphataemic rickets	IV moderate
Eralp et al. [24]	Turkey	Case report	Investigate the result of treatment with fixator-assisted intramedullary nailing in two cases with WSD.	Two patients with WSD, treated for vitamin-D-resistant rickets at younger age.	Vitamin-D-resistant rickets	IV high
Gigante et al. [25]	Italy	Case series	Evaluate temporary hemiepiphysiodesis in lower limb deformities in children with renal osteodystrophy (ROD).	One of the seven patients with ROD had WSD. Started as unilateral varus, developed valgus alignment in the contralateral knee.	ROD	IV high
Gupta et al. [26]	India	Literature review	Review the different types of nutritional vs. non-nutritional rickets.	Mention of WSD as a skeletal finding in nutritional rickets. It is not mentioned in non-nutritional rickets.	Nutritional rickets	n.a. *
Ikegawa [27]	Japan	Literature review	Review of the recent advances and current status of the genetic analysis of skeletal dysplasias.	Describes one WSD case, 17 years old with genu valgum on the left.	Skeletal dysplasia	n.a. *
Iyer and Diamond [28]	USA	Literature review	Review the effects of the resurgence of vitamin D deficiency and rickets.	Describes WSD as a possible clinical presentation of rickets.	Vitamin D deficiency rickets	n.a. *
Iyer and Diamond [29]	USA	Literature review	Review of the clinical, radiographic and biochemical manifestations of rickets.	Describes WSD as a possible clinical presentation of rickets.	Vitamin D deficiency rickets	n.a. *
Kenis et al. [30]	Austria	Case report	To describe the deformities in a patient with dysspondyloen-chondromatosis (DSC).	One patient with WSD with genu valgum of 30° on the right, genu varum of 10° on the left side. Age at start walking: 3 years.	DSC	IV high
Kim et al. [31]	Korea	Case series	Investigate the mutation frequency in individuals with multiple epiphyseal dysplasia (MED) and identify radiographic predictors.	Two of the fifty-five patients that identified with a previously reported mutation pathogenic for MED presented with WSD at the knee. One MATN3 and one COMP mutation.	MED	IV moderate
Lambert and Linglart [6]	France	Literature review	To describe the different causes and therapies of genetic and nutritional rickets.	WSD in walking children is a clinical manifestation of rickets.	Rickets	n.a. *
McKeand et al. [32]	USA	Case-control study	Describe the natural history of pseudoachondroplasia (PSACH).	WSD in 11/67 cases (16.4%), 8/11 cases (72.7%) needed a corrective operation for WSD.	PSACH	IIIb moderate
Muensterer et al. [33]	USA	Literature review	Describe pseudoachondroplasia, and its radiographic features.	In patients with pseudoachondroplasia, WSD typically develops around puberty, when genu varum transforms into WSD due to the progressive joint laxity.	PSACH	n.a.*
Nayak et al. [34]	India	Case report	Describe a case of epidermolytic hyperkeratosis (EHK) with rickets.	Epidermolytic hyperkeratosis (EHK) with rickets in a 6-year-old boy showed progressive WSD since age 3.	EHK	V high
Nishimura et al. [35]	Germany	Case series	Describe TRPV4 mutations in patients with spondylo-epiphyseal dysplasia (SED) and parastremmatic dysplasia.	A 7-year-old patient with reported low birth weight and length. Onset of walking at age 4. At 7 years she had a short stature (-4SD) and WSD. A TRPV4 mutation was found.	Parastremmatic dysplasia (TRPV4 mutation)	IV moderate
Nishimura et al. [36]	Switzerland	Literature review	To describe the different skeletal dysplasia’s related to TRPV4 mutations.	Patients with parastremmatic dysplasia have restricted joints and severe misalignment of the lower limbs (severe genu valgum, genu varum or WSD).	Parastremmatic dysplasia (TRPV4 mutation)	n.a. *
Oginni et al. [37]	Nigeria	Prospective case series	Response of oral calcium in Nigerian children with rickets.	Nine out of twenty-six children with underlying Ca-deficiency rickets presented with WSD, they were treated with calcium supplements, with good results.	Calcium-deficiency rickets	IV high
Oni and Keswani [38]	Nigeria	Case series	To describe the radiological findings of idiopathic or primary WSD.	Eight WSD patients were found, the onset of clinical and radiological alterations is abrupt, where the disease arises from a formerly normal epiphysis. The radiological features are similar to Blount, and therefore the etiological considerations that apply to Blount may also apply to primary WSD.	Hypotheses: similar to Blount, mechanical pressure, illness	IV moderate
Oni et al. [11]	Nigeria	Case series	To describe windswept deformity.	Eight patients with osteochondrosis with abrupt onset in previously healthy children, with formerly normal epiphyses.	Hypothesis: similar to Blount	IV moderate
Oyemade [13]	Nigeria	Case series	To describe the correction of primary knee deformities in children with and without rickets.	Rachitic patients WSD 12/47, Non-rachitic patients: WSD 15/67.	Rachitic or idiopathic (Blount)	IV moderate
Oyemade [1]	Nigeria	Case series	To clarify aetiological factors in primary deformities of the knee in children.	WSD: peak age male and female 2 years. Rachitic: WSD (12/47). Non-rachitic WSD (15/67).	Rachitic and non-rachitic Blount-like (weight-bearing)	IV moderate
Paruk et al. [39]	South Africa	Case report	Describe two cases of primary hyperparathyroidism (PHPT) in adolescence, mimicking rickets.	A 13-year-old male with progressive pain and WSD (right varus, left valgus). Caused by a parathyroid adenoma.	PHPT	IV high
Pavone et al. [40]	Italy	Literature review	Review hypophosphataemic rickets.	WSD described as a clinical feature of X-linked hypophosphataemic rickets.	X-linked hypophosphataemic rickets	n.a. *
Pettifor et al. [10]	South Africa	Literature review	Presentation of vitamin D deficiency and nutritional rickets in children.	In older children with vitamin D deficiency rickets, WSD may be present.	Vitamin D deficiency rickets and Calcium deficiency rickets	n.a. *
Pettifor et al. [41]	South Africa	Case series	Clinical, radiographic and biochemical findings in four children with severe bone deformities resembling rickets.	One out of four children had WSD.	Calcium-deficiency rickets	IV moderate
Prakash et al. [42]	India	Prospective cohort	To evaluate the behaviour of lower limb deformities due to rickets.	Five out of one-hundred and seventeen nutritional rickets patients had WSD. Varus deformity being the youngest, valgus and WSD being older.	Nutritional rickets	Iib high
Prentice et al. [43]	Gambia	Case-control study	Biochemical profile in Gambian children with rickets of unknown aetiology and normal 25OHD.	One out of thirty-seven patients had WSD.	Calcium-deficiency rickets	IIIb high
Shehzad and Shaheen [44]	Pakistan	Case report	Describe a case of epidermolytic hyperkeratosis (EHK) with rickets.	A 13-year old female with WSD, started around the age of 5. Scaling of the skin since birth.	Epidermolytic hyperkeratosis (EHK) with rickets	IV high
Simsek-Kiper et al. [45]	Turkey	Case series	Report on five patients from 2 unrelated families with SEMDFA (spondyloepimetaphyseal dysplasia Faden-Alkuraya type).	One patient presents with WSD (right genu varum, left genu valgum).	SEMDFA	IV High
Smyth [4]	Nigeria	Case report	Describe three cases of windswept deformity.	Three cases of WSD. Two cases with normal development, when suddenly WSD develops. In one case, there is a period of an acute febrile illness (possibly measles) preceding the development of WSD.	Period of epiphyseal instability + stress factor, geographical genetic dysplasia	IV moderate
Solagberu [12]	Nigeria	Prospective case series	Determine the varieties of angular deformities of the knee in children, in/around Ilorin, Nigeria.	Ten patients with WSD presented in one year. Age distribution between 2 and 5 years.	One bone diseased, while the other appears to be compensating	IV moderate
Teotia et al. [46]	India	Literature review	Report effects of endemic fluoride exposure on metabolic bone disease.	WSD as presentation of bony leg deformity due to high levels of fluoride exposure in drinking water.	Endemic chronic fluoride toxicity	n.a. *
Thacher et al. [47]	Nigeria/South Africa	Case report	Three cases of vitamin D-deficiency rickets associated with ichthyosis.	WSD in two patients with ichthyosis and rickets.	Ichthyosis with rickets	IV high
Thacher et al. [7]	Nigeria	Literature review	Describe the features of calcium-deficiency rickets.	Bowleg deformity is less specific for active rickets than knock-knee or WSD.	Calcium-deficiency rickets	n.a. *
Thacher et al. [9]	Nigeria	Case-control study	Determine whether low dietary calcium intake is associated with rickets in Nigerian children.	Of 123 Nigerian children with rickets: 16 (13%) had WSD.	Calcium-deficiency rickets	IIIb high
Thacher et al. [8]	Nigeria	Cohort study	Development of a clinical prediction model for active rickets.	The median age of onset of WSD was 24 months. WSD present in 39 of the 278 cases of active rickets (14%). Children presenting leg deformities over a span of 4 years, 95/736 had WSD (12.9%).	Rickets	IIb moderate
Vatanavicharn et al. [48]	USA	Case report	Radiographic patellar finding in a patient with pseudoachondroplasia (COMP mutation).	At age 5, the patient developed WSD (right genu varum, left valgum).	Pseudoachondroplasia (following COMP mutation)	IV high
Weiner et al. [49]	USA	Case series	Characterise the typical orthopaedic findings in pseudoachondroplasia.	Angular deformity of knees: genu valgum *n* = 35 (22%), genu varum *n* = 89 (56%), WSD *n* = 35 (22%). Laxity in all patients.	Pseudoachondroplasia (following COMP mutation)	IV moderate
Yilmaz et al. [50]	Croatia	Cohort study	Temporary hemi epiphysiodesis for correction of angular deformities in children with skeletal dysplasia.	One patient with metaphyseal dysplasia had WSD and underwent correction for both varus and valgus deformity.	Metaphyseal dysplasia	Iib moderate

* n.a.: not applicable, articles classified as “literature review” did not meet the criteria for level of evidence scoring or critical appraisal. Literature reviews did not provide the number of WSD patients included and therefore these numbers are not mentioned in the table for literature reviews.

In Table 2, a quantitative summary was made on the demographic data of windswept deformity. A total of 184 patients with windswept deformity were included in Table 2, in 69 cases the gender was described: 45 (65.2%) males and 24 (34.8%) females. Of the 170 patients where the race is described, 108 (63.5%) are of African descent, 11 (6.5%) Asian and 50 (29.4%) Caucasian (37 of the Caucasian patients were retrieved from the same article about pseudoachondroplasia) [49]. One person of Bedouin descent was classified as Middle Eastern (0.6%). Based on the articles found in this study, an overview of the identifying features of each aetiology of windswept deformity is presented in Table 3. Figure 2 shows a flowchart that can be used to find the possible cause of windswept deformity in a child.

### 3.1. Rickets and Other Metabolic Disorders

About half of the articles included reported rickets in patients with windswept deformity (*n* = 23). In five articles, accounting for a total of 68 patients, the rickets type was unspecified [1,5,6,8,13]. However, in three articles, for a total of eight patients, rickets was found to be (X-linked) hypophosphataemic, due to diminished reabsorption of phosphate in the kidneys [19,23,40]. Nutritional rickets due to vitamin D or calcium deficiency was found in another 12 articles, accounting for another 36 patients with windswept deformity [7,9,10,22,24,26,28,29,37,41,42,43]. In four patients, from three articles, rickets was caused by a skin disorder, namely epidermolytic ichthyosis, which is present at birth [34,44,47]. This adds up to 116 patients with windswept deformity, likely due to rickets of varying types.

A further four types of metabolic disorders were found with windswept deformity at presentation, accounting for an additional six cases. These include primary hyperparathyroidism (PHPT) (*n* = 1) [39], chronic fluoride toxicity (*n* = 1) [46], distal renal tubular acidosis (dRTA) (*n* = 1) [21] and renal osteodystrophy (ROD) (*n* = 3) [20,25].

### 3.2. Skeletal Dysplasia and Other Genetic Disorders

Six cases of windswept deformity were found in patients with different types of dysplasia’s: multiple epiphyseal dysplasia (MED) (*n* = 6) [31], dysspondyloenchondromatosis (DSC) (*n* = 1) [30], metaphyseal dysplasia (*n* = 1) [50], parastremmatic dysplasia (with TRPV4 mutation) (*n* = 1) [35,36] and spondyloepimetaphyseal dysplasia Faden-Alkuraya type (SEMDFA) (*n* = 1) [45]. These skeletal dysplasias are often characterised by a specific mutation. In MED cases, the mutations are either on the COMP or MATN3 gene, the former also being known to cause pseudoachondroplasia (PSACH), another disease in which windswept deformity can be found. We found 48 patients with PSACH presenting with windswept deformity, from five articles [27,32,33], 36 of which bring confirmed COMP mutations [48,49]. Schwartz-Jampel Syndrome (SJS) was another syndrome in which one patient with windswept deformity has been described [18]. Patients with skeletal dysplasia often present with scoliosis and consequently WSD towards the contralateral side.

### 3.3. Trauma

In one patient, trauma seemed to be the cause of the windswept deformity [14]. This patient was a young female with a history of fractures, often untreated due to congenital insensitivity to pain, presumably being the cause of the angular deformities of the legs. 

### 3.4. Descriptive Articles without Specific Underlying Disorder

We grouped six articles together with a total of 36 patients, in which different hypotheses for the cause of windswept deformity were brought forward: a combination of mechanical pressure and a period of illness (similar cause as Blount disease) (*n* = 8) [11,38]; excessive or early weight-bearing, dietic or ethnicity (*n* = 15) [1,13]; a combination of the previously mentioned factors (*n* = 3) [4]; compensation, where only one side is diseased, while the other side compensates (*n* = 10) [12]. All of these hypotheses were (partly) based on mechanical loading or weight-bearing. The patients in these articles were all African and typically presented with windswept deformity between the age of 2 and 3 years, being otherwise healthy. None of these patients showed signs of (healed) rickets.

## 4. Discussion

This systematic review gives an overview of all previously published aetiology hypotheses for windswept deformity. Windswept deformity is generally limited to an individual with genu valgum on one side, and genu varum on the contralateral side. Further specific phenotypical descriptions vary depending on the aetiology. 

Our results display that windswept deformity can be a manifestation of a broad variety of pathologies. However, in patients in whom no underlying illness was found, the deformity was deemed idiopathic, and the following hypotheses are brought forward: weight-bearing (due to excessive weight or early walking) [1,11,13,38], epiphyseal instability [4], stress factors (illness) [4] and geographical genetic factors [1,4,13]. Though these hypotheses are comparable between articles, they lack supporting evidence.

Rickets of different types was found to be the most common pathology manifesting windswept deformity. Although it appears to be the most frequent cause of windswept deformity, windswept deformity is far from the most common presentation of rickets. Thacher [7], presents different clinical features and their utility in predicting radiologically active rickets; only 14% of the children with radiologically active rickets have windswept deformity and the probability that a child with windswept deformity has radiologically active rickets is 41%. Unfortunately, no further research has been conducted on factors influencing the development of windswept deformity, genu valgum or genu varum in children with rickets. Bhimma et al. [22] concluded that mainly vitamin D deficiency is responsible for rickets, however, this may be aggravated by calcium deficiency. These two deficiencies may explain the geographical and ethnic distribution of windswept deformity. As seen in Table 2, most of the patients with windswept deformity were African, and about half of the included studies were performed in sub-Saharan African countries. The diet of rural African children is often low in milk and other dairy products, hence leading to a reduced calcium intake from the diet, despite the fortification of products with calcium [51]. The vitamin D deficiency may be explained by the increased sunlight exposure required for black children [22], or by cultural and/or religious factors that limit the exposure to sunlight, such as covering garments or veils. Hypophosphataemic rickets is most commonly found in its X-linked inheritance form and causes rickets due to its defects in renal handling of phosphorus [52]. Furthermore, epidermolytic ichthyosis is described in windswept patients caused by rickets due to marked hyperkeratosis of the skin [53]. Consequently, there is a decreased synthesis of vitamin D in the epidermis stimulating parathyroid hormone secretion, and a higher risk of rickets [54].

The cases of windswept deformity occurring in patients with other metabolic disorders are often comparable to the pathophysiology behind the different forms of rickets. In calcium deficiency rickets, patients present with secondary hyperparathyroidism as the low calcium levels stimulate the increased production of PTH. Similarly, primary hyperparathyroidism, distal renal tubular acidosis (dRTA) and renal osteodystrophy cause abnormalities in calcium and phosphorus levels. Exposure to high levels of fluoride in drinking water may lead to a decrease in strength by altering the structural integrity of the bone microarchitecture, which possibly leads to skeletal deformities, such as windswept deformity. Teotia et al. describe the difficulty in differentiating calcium-deficiency rickets from fluoride toxicity, and believe that every child presenting with bone disease in areas endemic to fluorosis is a case of skeletal fluorosis until proven otherwise [46].

A variety of skeletal dysplasias have been described in cases of windswept deformity. Often, these dysplasias are caused by a specific mutation, and a positive family history may therefore be present. Additionally, these patients often present additional symptoms, for example, brachydactyly and craniosynostosis for SEMDFA [45]. PSACH is caused by a mutation in the COMP gene and is usually inherited in an autosomal dominant manner. Although PSACH is not usually discovered until the age of 2-3 years, when disproportionate short stature, waddling gait and evidence of increased joint laxity starts to develop, these features should be used to distinguish PSACH as the cause of windswept deformity [49]. Most patients with a syndrome or dysplasia presenting with windswept deformity have other clinical features, such as malformations of the eyes and face, which can be used to identify the underlying syndrome.

Only a single article, by Bar-On et al. [14], describes trauma leading to windswept deformity. The precise location of the fractures is not described, and therefore, it is impossible to conclude whether this case of windswept deformity was due to compensation for one malformed leg, or if the trauma occurred in both legs, leading to the deformity. The child in this single case suffered from congenital insensitivity to pain, and therefore had a higher risk of multi-trauma leading to lower limb deformities.

The remaining articles, in which other hypotheses for windswept deformity were described, explained its development in patients where there is no underlying cause found. We found no explicit evidence of early walking or excessive weight. Although the beforementioned causes were poorly reported, all patients in these studies had comparable stories. All were of African descent, otherwise healthy, with no signs of healed rickets and the age of onset was between 2 and 3 years of age. There might be geographical or genetic factors that could explain the distribution of windswept deformity in the unspecified group.

When a child presents with windswept deformity, the number of possible underlying causes is extensive, and a complete overview is helpful to make the correct diagnosis. Table 3 shows an overview of the identifying features of each aetiology found in this review. To find a possible cause we advise extensive history taking, including family history (to exclude genetic dysplasias or syndromes) and developmental history; detailed physical examination, looking for clinical features suggestive of rickets (thickened wrists and ankles, and/or rickety rosary), or other identifying features associated with dysplasias or syndromes (facial dysmorphisms, ligamentous laxity, deformities at multiple joints, etc.); and additional testing, such as X-rays of the lower limb and wrist, looking for evidence of rickets or other abnormalities which might fit specific skeletal dysplasias, blood panel (alkaline phosphatase, calcium, phosphate, magnesium, PTH, 25-OH-vitamin D, albumin, creatinine and fluoride) and spot urine test (calcium, phosphate, creatinine and Ph) to exclude or identify rickets and/or other metabolic causes. If required, additional blood tests (e.g., chloride, potassium, 1.25-di-OH-vitamin D, arterial blood gas (ABG) and FGF23) can be performed. Genetic analysis may be indicated when a specific dysplasia or syndrome (COMP, MATN3, TRP4, RSPRY-1 and HSPG2 mutation), or hypophosphataemic rickets (PHEX mutation) is suspected. Additionally, a renal ultrasound can be performed in the case of distal renal tubular acidosis or a skin biopsy to confirm hyperkeratosis. However, when the cause of windswept deformity cannot be found it may be multifactorial, including mechanical loading or weight-bearing.

Despite the large number of articles included in this review, only very few focus on the aetiology of windswept deformity, and most articles have a different aim, describing the deformity as an ancillary finding. Additionally, most articles have old publishing dates. On the other hand, most articles occur in low and middle-income countries, which may result in an underestimation of the problem as less research is conducted in low and middle-income countries. Hence, the specific data available on patients with windswept deformity is often limited. Despite not being found in the literature as causes of windswept deformity, logically there are more underlying disorders that can cause windswept deformity, for instance, other skeletal dysplasias or metabolic disorders (e.g., hypomagnesemia or hypo-albuminemia).

## 5. Conclusions

Currently, the existing evidence on the aetiology of windswept deformity of the knee shows a broad spectrum of underlying causes that may lead to its development. However, when none of these specific causes can be identified, it appears that the aetiology is multifactorial, resting on the hypotheses of weight-bearing, epiphyseal instability, stress factors and geographical genetic factors. Presently, not enough evidence is available to confirm these hypotheses, and more research is necessary. Nevertheless, we have presented an overview, which helps guide clinicians presented with a case of windswept deformity. A thorough (family and developmental) history, followed by physical examination, and additional tests, such as X-rays, blood panels, urine tests, renal ultrasound and genetic analysis, may help identify the underlying disorder.

## Figures and Tables

**Figure 2 children-09-00703-f002:**
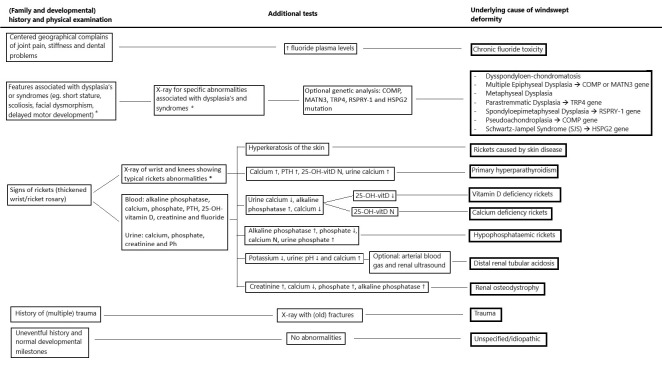
Flowchart to find the possible cause of windswept deformity in a child. * See Table 3 for more detailed descriptions.

**Table 2 children-09-00703-t002:** Quantitative overview of the demographic data from the studies describing the aetiology windswept deformity *.

Author	Country of Study	WSD Cases	Male	Female	Suggested Aetiology or Hypothesis	Ethnicity	Onset Age (Months)	Onset Valgus	Onset Varus	Valgus Right	Valgus Left
Akpede et al. [5]	Nigeria	2	-	-	Biochemical rickets	African	-	-	-	-	-
Al Kaissi et al. [49]	Austria	1	1	0	Schwarz-Jampel Syndrome (SJS)	-	-	-	-	-	1
Al Kaissi et al. [19]	Austria	7	7	0	X-linked hypophosphataemic rickets	-	14–18	-	-	-	-
Bar-On et al. [20]	Israel	1	1	0	Renal osteodystrophy (ROD)	-	-	1	-	1	-
Bar-On et al. [14]	Israel	1	0	1	Congenital insensitivity to pain (trauma)	Middle-East	-	-	-	1	-
Bharani et al. [21]	India	2	2	0	Distal renal tubular acidosis (dRTA)	Asian	36	-	-	2	-
Bhimma et al. [22]	South Africa	3	-	-	Vitamin D and/or Ca deficiency rickets	African	-	-	-	-	-
Dudkiewicz et al. [23]	Israel	1	1	0	Hypophosphataemic rickets	-	-	-	-	1	-
Eralp et al. [24]	Turkey	2	1	1	Vitamin D-resistant rickets	-	-	-	-	2	-
Gigante et al. [25]	Italy	1	1	0	Renal osteodystrophy (ROD)	-	-	-	1	1	-
Ikegawa [27]	Japan	1	1	0	Pseudoachondroplasia	Asian	-	-	-	-	1
Kenis et al. [30]	Austria	1	1	0	Dysspondyloenchondromatosis (DSC)	Caucasian	-	-	-	1	-
Kim et al. [31]	Korea	2	2	0	Multiple epiphyseal dysplasia (MED)	Asian	-	-	-	-	-
McKeand et al. [32]	USA	11	-	-	Pseudoachondroplasia	Caucasian	-	-	-	-	-
Nayak et al. [34]	India	1	1	0	Rickets (epidermolytic hyperkeratosis)	-	36	-	-	-	-
Nishimura et al. [35]	Germany	1	0	1	Parastremmatic dysplasia (with TRPV4 mutation)	African	-	-	-	-	1
Oginni et al. [37]	Nigeria	9	-	-	Ca deficiency rickets	African	-	-	-	-	-
Oni and Keswani [38]	Nigeria	8	3	5	Similar to Blount mechanical pressure + illness	African	6–24	1	1	2	-
Oyemade [1]	Nigeria	28	18	10	Rickets (12) or non-rachitic (15) (3 hypotheses: weight-bearing, dietetic, genetic)	African	12–108	-	-	-	-
Paruk et al. [39]	South Africa	1	1	0	Primary hyperparathyroidism	African	150	-	-	-	1
Pettifor et al. [41]	South Africa	1	0	1	Ca deficiency rickets	African	-	-	-	1	-
Prakash et al. [42]	India	5	-	-	Nutritional rickets	Asian	48–120	-	-	-	-
Prentice et al. [43]	Gambia	1	-	-	Ca deficiency rickets	African	-	-	-	-	-
Shehzad and Shaheen [44]	Pakistan	1	0	1	Rickets (epidermolytic hyperkeratosis)	Asian	60	-	-	-	-
Simsek-Kiper et al. [45]	Turkey	1	1	0	Spondyloepimetaphyseal dysplasia Faden-Alkuraya type (SEMDFA)	Caucasian	36–48	-	-	-	1
Smyth [4]	Nigeria	3	2	1	Period of epiphyseal instability + stress factor, geographical genetics	African	24–54	3	-	1	1
Solagberu [12]	Nigeria	10	-	-	Compensation for 1 diseased bone	African	24–60	-	-	-	-
Thacher et al. [47]	Nigeria/South Africa	2	1	1	Ichthyosis with rickets	African	-	-	-	-	-
Thacher et al. [8]	Nigeria	39	-	-	Rickets	African	-	-	-	-	-
Vatanavicharn et al. [48]	USA	1	0	1	Pseudoachondroplasia	Caucasian	60	-	-	-	1
Weiner et al. [49]	USA	35	-	-	Pseudoachondroplasia	Caucasian	-	-	-	1	-
Yilmaz et al. [50]	Croatia	1	0	1	Metaphyseal dysplasia	Caucasian	-	-	-	-	1
Total		184	45	24		African *n* = 108 Asian *n* = 11 Caucasian *n* = 50 Middle-east *n* = 1 Missing ethnicity *n* = 14		5	2	14	8

* All literature reviews and other overlapping study populations were excluded from this quantitative overview.

**Table 3 children-09-00703-t003:** An overview of the clinical presentation of each specific aetiology.

Aetiology Group	Specific Aetiology	Family History	History	Physical Exam	Laboratory Findings	X-Ray	Articles
**Rickets**	Vitamin D deficiency rickets*	Generally uneventful	Decreased exposure to sunlight	Thickened wrists and ankles, rickety rosarymuscle weakness	Alkaline phosphatase ↑ Phosphate ↓/N/↑ Calcium ↓ 25-OH-vitD ↓ 1.25-di-OH-vit D ↓/N/↑ PTH ↑ Urine: Ca ↓	Widening of the growth plate and abnormal configuration of the metaphysis: – Fraying indistinct margins of the metaphysis – Splaying: widening of metaphyseal ends – Cupping: concavity of metaphysis Most seen in the wrist and knee Anterior rib ends: rachitic rosary	Bhimma et al. [22], Thacher [7], Iyer et al. [28,29], Gupta et al. [26], Prakash et al. [42]
	Calcium deficiency rickets*	Generally uneventful	Low-calcium diet	Thickened wrists and ankles, rickety rosary	Alkaline phosphatase ↑ Phosphate ↓ Calcium ↓ 25-OH-vitD N 1.25-di-OH-vit D ↑ PTH ↑ Urine: Ca ↓	See above	Bhimma et al. [22], Oginni et al. [37], Prentice et al. [43], Pettifor et al. [41], Gupta et al. [26]
	Hypophosphataemic rickets	X-linked (PHEX mutation) or autosomal dominant (FGF23 mutation) transmission	Delayed walking, muscular weakness, bone pain, failure to thrive, tooth abscesses	Thickened wrists and ankles, rickety rosary, dental abnormalities	Alkaline phosphatase ↑ Phosphate ↓ Calcium N 25-OH-vitD N 1.25-di-OH-vitD N/↓ PTH N/↑ FGF23 ↑Urine: phosphate ↑ Genetic testing: PHEX mutation	See above	Bhimma et al. [22], Al-Kaissi et al. [19], Dudkiewicz et al. [23], Pavone et al. [40], Eralp et al. [24], Gupta et al. [26], Prakash et al. [42]
	Skin disease (Epidermolytic hyperkeratosis or Ichthyosis)	Consanguinity and familial inheritance may occur	Bright red blisters after birth. Development of hyperkeratotic plaques	Generalised dry skin, hyperkeratotic and cobble-stone plaques. Rib beading, widening of wrists and ankles	Alkaline phosphatase ↑ Phosphate ↓ Calcium ↓ 25-OH-vitD ↓ PTH ↑ Skin biopsy: hyperkeratosis	See above	Shehzad and Shaheen [44], Nayak et al. [34], Thacher [47]
**Other metabolic**	Primary hyperparathyroidism	Generally uneventful	Progressive pain, normal developmental milestones	No abnormalities	Alkaline phosphatase ↑ Phosphate ↓ Calcium ↑ PTH ↑ 25-OH-vitD N 1.25-di-OH-vitD ↑ Urine: Ca ↑	See above and a sestamibi scan: increased focal uptake of the parathyroid glands	Paruk et al. [39]
	Chronic fluoride toxicity	Affected family members, centred geographic distribution of fluoride levels	Mild: generalised bone and joint painmoderate: stiffness and rigiditySevere: flexion deformities at hips and knees	Stiff and rigid spine and joints, flexion deformity hips, knees and elbows, hypo-mineralisation of tooth enamel	Alkaline phosphatase ↑ Phosphate N Calcium N 25-OHD N1-25(OH)2D ↑ PTH ↑ Plasma fluoride ↑	Osteosclerosis, periosteal bone formation, calcifications of interosseus membrane, rickets-like metaphyses	Teotia et al. [46]
	Distal renal tubular acidosis	Familial inheritance	Sickle cell disease, failure to thrive, polyuria, polydipsia	Low weight/height, frontal bossing, wrist widening (signs of rickets)	ABG: Metabolic acidosis Potassium ↓ Chloride ↑ Urine: pH↓, calcium ↑ Renal ultrasound: nephrocalcinosis	Osteopenia, angular deformities and signs of rickets	Bharani et al. [21]
	Renal Osteodystrophy	Familial inheritance may occur	Bone pain, muscle weakness	Significant growth retardation	Alkaline phosphatase ↑ Phosphate ↑ Calcium ↓ Creatinine ↑ PTH ↑	Widening and elongation of the growth plates and cupping of the metaphysis and signs of rickets	Bar-On et al. [20], Gigante et al. [25]
**Dysplasia’s and syndromes**	Dysspondyloen-chondromatosis	Generally uneventful	Delays in motor development	Neonatal dwarfism, unequal limb length, flat midface with frontal prominence and progressive kyphoscoliosis	No abnormalities	Aniosospondyly and enchondroma-like lesions in the metaphyseal and diaphyseal portions of the long tubular bones	Kenis et al. [30]
	Multiple Epiphyseal Dysplasia	Familial inheritance may occur	Joint pain, scoliosis, deformities hands, feet, knees and hips	Muscular hypotonia, ligamentous hyperlaxity, abnormal gait, angular deformities at hips and knees	Genetic testing: COMP or MATN3 mutations	COMP: small and round femoral head, MATN3: crescent-shaped femoral head	Kim et al. [31]
	Metaphyseal Dysplasia	Consanguinity and familial inheritance may occur	Mental, physical and height development are usually normal	Angular deformities of the knees, palpable widening of the distal femur and clavicles	no abnormalities	Erlenmeyer flask deformity	Yilmaz et al. [50]
	Parastremmatic Dysplasia	Generally uneventful	Normal mental milestones, motor development may be slightly delayed, short stature	Windswept and flexural deformity of the legs, scoliosis, platyspondyly	Genetic testing: TRP4 mutation	Flaky metaphyses with wide zones of radiolucencies and flocky calcifications, disorganised epiphyseal ossifications, severe platyspondyly	Nishimura et al. [35,36]
	Spondyloepimetaphyseal Dysplasia Faden-Alkuraya type	Parental consanguinity, autosomal recessive inheritance	Difficulty walking, short stature, delayed motor and mental development	Short stature, hypertelorism, brachycephaly, short nose with depressed nasal bridge, tented upper lip, proptosis	Genetic testing: RSPRY-1 mutation	Mild spondylar dysplasia, epi-metaphyseal dysplasia of long bones (flat and irregular epi- and metaphyseal flaring)	Siimsek-kiper et al. [45]
	Pseudoachondroplasia	Autosomal dominant inheritance	Normal birthweight and length, At around 2-4yr of age short stature and disproportionately short limbs appear	Short stature, disproportionately short limbs, short and stubby fingers, increased joint laxity, waddling gait	Genetic testing: COMP mutation	Irregular or fragmented epiphyses, flaring, widening or trumpeting of the metaphyses, anterior beaking of vertebrae	Muensterer et al. [33], Vatanavicharn et al. [27], Weiner et al. [49], McKeand et al. [32]
	Schwartz-Jampel Syndrome (SJS)	Parental consanguinity and familial inheritance	Normal gestation with severe muscle stiffness at birth	Dysmorphic facial features, trismus	Genetic: HSPG2 gene mutation	Kyphoscoliosis, platyspondyly with coronal clefts in vertebrae, inferior femoral and superior tibial epiphyses look enlarged and distorted	Al-Kaissi et al. [48]
**Trauma**	Trauma	Generally uneventful	History of fractures	Abnormal gait, signs of bruises and evidence of (old) fractures	Non specific	Evidence of (old) fractures	Bar-On et al. [14]

* Lab findings in vitamin D deficiency and hypocalcemia rickets depend on the phase of rickets. Phase 1: hypocalcemia causes PTH rise, leading to bone resorption and hyperphosphatemia and rise in alkaline phosphatase. This phase has a relative resistance to PTH. Phase 2: PTH rises further and overcomes the resistance. Calcium rises to normal or slightly lower than normal range and phosphate decreases further (renal excretion due to PTH). Phase 3: hypocalcemia returns worse because of depleted reserves, hypophosphatemia persists and further rise of alkaline phosphatase. In phase 3 X-ray abnormalities become visible. ↓ : lower compared to reference standard. ↑ : higher compared to reference standard.

## Data Availability

Not applicable.

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
