# Peer review of "Windswept Deformity a Disease or a Symptom? A Systematic Review on the Aetiologies and Hypotheses of Simultaneous Genu Valgum and Varum in Children"

_children, 2022, doi:10.3390/children9050703_

Round 1
Reviewer 1 Report
The paper is very well written, in fluent scientific English and complete. It provides an interesting overview of the literature on windswept deformity, aiming to support the clinicians and physicians during clinical practice. For example, table 2, in my opinion is a useful tool.
I suggested to add few minor integrations to align the manuscript with PRISMA. Moreover, I have some minor comments/hints to implement the manuscript; particularly the tables, that are crucial in systematic review.
Minor revisions:
- Section 2.1 eligibility criteria: please specify the age in number (not using only the term childhood), but indicating the min and max age considered
- Section 2.2 information source: move the date of sources consultation from results to this section
- Section 2.3: rename from “search” to “search strategies” or “search actions” or similar
- Section 3: please give some additional information/details on the 6 references that seems to meet eligibility and then were excluded, according to PRISMA checklist.
- Table 1: what does it mean the n.a.* (review)? Please specify in the table footnote
- Table 1: from Aetiology WSD to WSD aetiology
- Table 1: harmonize the content. Put the dot period in all sentences or remove from all
- Table 1: please organize a little better the table to improve readability. In some articles, texts from aim of study, elaboration of WSD and/or aetiology are too close (like in Nishimura et al., Oginni et al, etc)
- Table 1: in some studies (like Teotia or Weiner) the number of patients with WSD is missing. Please specify
- Table 2: in the * note correct the sentence. ReviewS and studyS
- Table 2: change race with ethnic origin or ethnicity
- Table 2: add the missings in the total. Missing ethnic origin n=14
- Results: erroneous numbering of sections
- Results line 168: Rephrase “In MED cases the mutations are either on COMP or MATN3 gene,..”
- Results line 17: “36 of which bring confirmed”
- Results line 183: since you dedicated a section for genetic disorders, it’s unclear why you mention the genetic factors in this section. Please explain.
- Table 3: harmonize the content. Put the dot period in all sentences or remove from all
- Table 3: please organize a little better the table to improve readability.
- Results line 193: and, not And
Author Response
First of all, thank you for the valuable feedback. I provided the list with possible comments down below.
- Section 2.1 eligibility criteria: please specify the age in number (not using only the term childhood), but indicating the min and max age considered
- Changes made
- Section 2.2 information source: move the date of sources consultation from results to this section
- Changes made
- Section 2.3: rename from “search” to “search strategies” or “search actions” or similar
- Changes made
- Section 3: please give some additional information/details on the 6 references that seems to meet eligibility and then were excluded, according to PRISMA checklist.
- Changes made
- Table 1: what does it mean the n.a.* (review)? Please specify in the table footnote
- Changes made
- Table 1: from Aetiology WSD to WSD aetiology
- Changes made
- Table 1: harmonize the content. Put the dot period in all sentences or remove from all
- Changes made
- Table 1: please organize a little better the table to improve readability. In some articles, texts from aim of study, elaboration of WSD and/or aetiology are too close (like in Nishimura et al., Oginni et al, etc)
- Changes made
- Table 1: in some studies (like Teotia or Weiner) the number of patients with WSD is missing. Please specify
- For most the changes are made, but for all literature reviews included no clear number of patients is stated in the studies and therefore not mentioned in the table.
- Table 2: in the * note correct the sentence. ReviewS and studyS
- 'Reviews' is corrected. The 'study' in this sentence is part of 'study participants' therefore not corrected.
- Table 2: change race with ethnic origin or ethnicity
- Changes made
- Table 2: add the missings in the total. Missing ethnic origin n=14
- Changes made
- Results: erroneous numbering of sections
- Changes made
- Results line 168: Rephrase “In MED cases the mutations are either on COMP or MATN3 gene,..”
- Changes made
- Results line 17: “36 of which bring confirmed”
- Changes made
- Results line 183: since you dedicated a section for genetic disorders, it’s unclear why you mention the genetic factors in this section. Please explain.
- Genetics changed for ethnicity
- Table 3: harmonize the content. Put the dot period in all sentences or remove from all
- Changes made
- Table 3: please organize a little better the table to improve readability.
- Changes made
- Results line 193: and, not And
- Changes made
Reviewer 2 Report
The Paper is a systematic review with the aim to expose the aetiologies for windswept deformity. The Authors deliver a well written and methodologically correct systematic review about this matter that explains the main aetiologies and gives an important overview for the diagnosis like stated in the discussion. Unfortunately the rarity of the condition and the extremely wide manifestation didn't help for better results.
Furthermore, I praise the Authors effort on using the Joanna Briggs Institute tool for assess the quality of the papers.
Author Response
Thank you for your kind feedback.
Reviewer 3 Report
Dear authors,
thank you for submitting your manuscript to Children.
The manuscript is interesting and good work.
Title: good
Abstract: good
Introduction: good
Methods: adequate and well presented
Results: fine
Discussion:good
Conclusion: adequate
No major corrections needed.
Author Response
Thank you for your kind feedback!